# Mining of Minor Disease Resistance Genes in *V. vinifera* Grapes Based on Transcriptome

**DOI:** 10.3390/ijms242015311

**Published:** 2023-10-18

**Authors:** Junli Liu, Liang Wang, Shan Jiang, Zhilei Wang, Hua Li, Hua Wang

**Affiliations:** 1College of Enology, Northwest A&F University, Xianyang 712100, China; liujunly@nwafu.edu.cn (J.L.); wangliang1999@nwafu.edu.cn (L.W.); jiangshan@nwafu.edu.cn (S.J.); wangzhilei@nwafu.edu.cn (Z.W.); 2China Wine Industry Technology Institute, Yinchuan 750021, China; 3Shaanxi Engineering Research Center for Viti-Viniculture, Xianyang 712100, China; 4Engineering Research Center for Viti-Viniculture, National Forestry and Grassland Administration, Xianyang 712100, China

**Keywords:** *V. vinifera*, disease resistance, microactive polygenes, transcriptome

## Abstract

Intraspecific recurrent selection in *V. vinifera* is an effective method for grape breeding with high quality and disease resistance. The core theory of this method is the substitution accumulation of multi-genes with low disease resistance. The discovery of multi-genes for disease resistance in *V. vinifera* may provide a molecular basis for breeding for disease resistance in *V. vinifera.* In this study, resistance to downy mildew was identified, and genetic analysis was carried out in the intraspecific crossing population of *V. vinifera* (*Ecolly* × *Dunkelfelder*) to screen immune, highly resistant and disease-resistant plant samples; transcriptome sequencing and differential expression analysis were performed using high-throughput sequencing. The results showed that there were 546 differential genes (194 up-regulated and 352 down-regulated) in the immune group compared to the highly resistant group, and 199 differential genes (50 up-regulated and 149 down-regulated) in the highly resistant group compared to the resistant group, there were 103 differential genes (54 up-regulated and 49 down-regulated) in the immune group compared to the resistant group. KEGG analysis of differentially expressed genes in the immune versus high-resistance group. The pathway is mainly concentrated in phenylpropanoid biosynthesis, starch and sucrose metabolism, MAPK signaling pathway–plant, carotenoid biosyn-thesis and isoquinoline alkaloid biosynthesis. The differential gene functions of immune and resistant, high-resistant and resistant combinations were mainly enriched in plant–pathogen interaction pathway. Through the analysis of disease resistance-related genes in each pathway, the potential minor resistance genes in *V. vinifera* were mined, and the accumulation of minor resistance genes was analyzed from the molecular level.

## 1. Introduction

*Vitis vinifera* (*V. vinifera*) has good quality and important application value in wine-making and fresh food [1], but various biotic and abiotic stresses have a significant influence on its yield and quality during its growth. Biological stress caused by pathogens is particularly important, which not only reduces fruit quality but also affects wine quality [2]. Traditional measures to prevent disease, such as spraying pesticides, not only increase production costs, reduce fruit quality and safety, and pollute the environment and soil but also lead to the accumulation of pesticides and their by-products [3,4]. Therefore, it is extremely important to obtain disease-resistant varieties without damaging fruit quality [5,6]. The disease resistance of Vitis species mainly exists in the wild grape of *V. rotundifolia* Michx and *V. amurensis*, but these populations tend to have lower yields and produce wines with suboptimal sensory characteristics [7]. The traditional methods of breeding for disease resistance are conducted with simple crossings and multi-generation recurrent crossings [8]. These two methods only focus on the utilization of major genes in *V. labrusca* or *V. amurensis* but neglect the improvement of parents in the breeding process; the fruit quality of intraspecific hybrids is more favorable to disease-resistant parents and is not suitable for large-scale extension. It is not suitable for the breeding of disease-resistant varieties of the minor resistance genotypes of the Eurasian species. Although *V. vinifera* showed susceptibility to all fungal diseases, there were differences in susceptibility among varieties. The reason for this difference was that there were multiple genes with minor resistance in *V. vinifera*; these genes can be stably inherited. Using the method of intraspecific recurrent selection in *V. vinifera*, the disease resistance of breeding materials could be improved continuously, and the new variety with good quality and disease resistance [9] could be bred on the premise of guaranteeing the fruit quality.

When plants are infected with pathogens, various defense mechanisms in plants are subsequently activated at the molecular level [10,11]. Many primary and secondary metabolite plants change when disturbed. Secondary metabolites are generally considered to be non-essential for basic plant metabolism. However, they not only provide energy when pathogens invade the recipient but are also involved in regulating the plant defense response in the presence of potential pathogens [12]. The secondary metabolites in model plant *Arabidopsis thaliana* and rice, including terpenoids, phenylpropanoids and alkaloids, are involved in the mechanism of plant self-protection [13]; these responses are controlled by signal transduction pathways [14]. Plant–pathogen interactions induce signal transmission, and innate immunity mainly includes pathogen-associated molecular patterns-triggered immunity (PTI) and effector-triggered immunity (Eti) [15]; signal transmission of these pathways needs further validation. The results showed that the glucose content and the ratio of glucose and fructose to sucrose were related to the disease resistance [16]. The immune-related hierarchical transcriptional response also clearly indicates that a complex regulatory circuit exists, including transcriptional activators and repressors [17]. Transcriptome sequencing has been widely used in plant disease resistance research, but the results are based on the signal pathway and related gene expression of plant response to pathogen infection. PCR and sequencing were used to analyze the resistance of *Botrytis cinerea*. It was found that the resistance of *Botrytis cinerea* to pyrimidine was relatively common, and the resistance to dimethomorph was still in the primary stage [18]. Two stable QTLs associated with resistance to *Botrytis cinerea* were detected by quantitative trait loci (QTLs) mapping based on two populations of hybrids (*Vitis riparia* × *Vitis labrusca*), which showed extensive segregation of resistance to *Botrytis cinerea* [19], combined with RNA-seq analysis, the structural gene VIEDR2 (Vitvi02g00982) and three transcription factors may be involved in *Botrytis cinerea* resistance [20]. Lei found that plant–pathogen interactions, phenylpropanoid biosynthesis, and metabolism are all related to grape-pathogen interactions during their transcriptome analysis of the berries of a species resistant to grape mature rot [21].

In order to make full use of the multi-genes of disease resistance in *V. vinifera*, a new variety with good quality and disease resistance was selected. We used the previously obtained disease-resistant variety *Ecolly* as the breeding intermediate material, conducted the intraspecific hybridization and the resistance identification, as well as the genetic analysis to downy mildew of the hybrid offspring. The plants with different resistance types were selected for transcriptomic analysis under natural conditions, starting from the pathways of secondary metabolite, signal transduction and environmental adaptation, carbohydrate and transcription factors, to explore the expression differences of multi-genes with minimal resistance in *V. vinifera*, and provide the molecular theory reference for the breeding of high-quality and disease-resistant *V. vinifera* varieties.

## 2. Results

### 2.1. Identification and Genetic Analysis of Disease Resistance

The average temperature in July, August and September of 2021 and 2022 in Yangling is concentrated between 20 and 30 °C, but there was a big difference in rainfall between the two years; therefore, there were significant differences in the disease index between different years. Genetic analysis was carried out on the hybrid parents *Ecolly* and *Dunkelfelder* and their progenies using the disease index of downy mildew. It was found that there was a difference in disease resistance between the hybrid parents *Ecolly* and *Dunkelfelder* in different years; the disease index in 2022 was lower than that in 2021 in both parents and *Dunkelfelder* > *Ecolly* (Figure 1a).

Genetic analysis of the disease index and resistance grade of two years hybrid population showed that the disease index of the hybrid progenies was normal distribution in vitro, and the quantitative traits were controlled by multiple genes (Figure 2b,c). In this population, due to significant differences in rainfall between 2021 and 2022, the disease index in 2021 is significantly higher than that in 2022. The overall resistance level of hybrid offspring in different years to downy mildew is relatively high, mainly distributed in disease resistance (disease index 5.1–25.0), high resistance (0.1–5.0), and immunity (0). Therefore, in subsequent transcriptomics experiments, representative plant samples were selected from these three levels for two consecutive years for analysis.

### 2.2. Transcriptome Sequencing Data Analysis

The result of detection and electrophoresis of 6 grape-treated samples showed that the RNA integrity of all samples was good and accorded with the requirement of sequencing, which could be used in follow-up experiments. As shown in Table 1, a total of clean data for all samples was obtained from the cDNA library, with clean data greater than 7G per sample. The proportion of Q20 and Q30 bases were more than 96.8% and 90%, respectively, and the GC content was between 45% and 47%. A large number of FM indexes were used to cover the whole genome, and the reads detected by RNA-seq were quickly and accurately compared with the genome by HISAT2; it was found that more than 87% of the sequenced reads were successfully aligned to the genome. In order to determine the reliability of the samples, the correlation analysis of the biological replicates of each group was carried out. The results showed that the correlations among the replicates were all R^2^ ≥ 0.98, which further indicated that the transcriptome sequencing quality was high and the samples were reliable and available for follow-up analysis.

### 2.3. Overall Analysis of Differential Genes

By analyzing the distribution and characteristics of the expression quantities of FPKM across samples (Figure 2a), it was found that more than 50% of the LOG10(FPKM) genes in all samples were concentrated between 0 and 2, with the mean log (FPKM) at approximately position 1. The gene expression values (FPKM) of all samples were analyzed by PCA (Figure 2b). The samples were divided into three groups according to their resistance characteristics and degree of dispersion: Immune (E: S1022, S1023), highly resistant (F: S1024, S1035) and resistant (G: S1031, S1037). An overall analysis of up-regulated and down-regulated DEGs was performed based on temperature variables, with the up-regulated and down-regulated genes classified by log2fold-change > 1 and log2fold-change < −1, and FDR < 0.05. Pairwise alignment of the three sets of transcripts gives a picture of gene expression differences in grape plants with different resistances (Figure 2c). The total number of differential genes between Group E and Group F (EvsF) was 546 (194 up-regulated and 352 down-regulated), Group E and Group G (EvsG) had a total of 199 (50 up-regulated and 149 down-regulated), and Group F and Group G (FvsG) had a total of 103 (54 up-regulated and 49 down-regulated). Under the same culture conditions, there were significant differences in gene expression patterns between the immune group and the high-resistance group plants. The plants in the high resistance and disease resistance groups have the least number of differential genes.

### 2.4. Differential Gene Analysis of Different Plant Samples

#### 2.4.1. GO Enrichment Analysis

The DEGs of transcriptome samples from different disease-resistant plants in EvsF, EvsG and FvsG combinations were analyzed for GO enrichment and distributed into three major categories (Figure 3). The distribution of differential genes among the three combinations was similar, and the enrichment of the GO term was consistent. In the category of Biological Process (Figure 3a), the differential genes of the three combinations are all enriched in the two GO terms of cellular process and metabolic process, followed by biological regulation, response to stimuli, and regulation of biological process. In the molecular function category(Figure 3b), in the three comparison groups, the differentially expressed genes were mainly enriched in the two GO terms of catalytic activity and binding and in the two GO terms of transporter activity and ATP-dependent activity. The three comparison groups had only two GO terms in the cellular component (Figure 3b) category: the cellular anatomical entity and the protein-containing complex.

The EvsF combination enriched the largest number of differential genes and further analyzed the top 20 enriched to GO term’s FDR value. The results indicate that adenosine monophosphate binding (GO: 0032559), adenosine monophosphate binding (GO: 0030554), ATP binding (GO: 0005524), protein kinase activity (GO: 0004672) and cell cycle (GO: 0007049) were the top five pathways with the highest number of differential genes. The first five pathways with the highest enrichment coefficients were microtubule binding (GO: 0008017), movement of cellular or subcellular components (GO: 0006928), microtubule-based movement (GO: 0007018), tubulin binding (GO: 0015631) and microtubule-based processes (GO: 0007017). Microtubule- and nucleotide-binding reactions played an important role in plant disease resistance. In addition, cytoskeleton protein binding (GO: 008092), cell cycle process (GO: 0022402), microtubule cytoskeleton (GO: 0015630), mitotic cell cycle (GO: 0000278), mitotic cell cycle process (GO: 1903047), cytoskeleton (GO: 0005856), microtubule (GO: 0005874), supramolecular polymer (GO: 0099081), supramolecular fiber (GO: 0099512) and polymeric cytoskeleton fiber (GO: 0099513) were also enriched.

Further analysis of the first 20 FDR values enriched to GO term for the EvsG group. The results found that the top five pathways with the highest number of differential genes were anion coordination (GO: 0043168), carbohydrate derivative binding (GO: 0097367), nucleotide binding (GO: 000166), nucleotide phosphate binding (GO: 1901265) and ribonucleotide binding (GO: 0032553). Adenosine monophosphate binding (GO: 0030554), adenine ribonucleotide binding (GO: 0032559), ADP binding (GO: 0043531), purine ribonucleotide binding (GO: 0017076) and purine ribonucleotide binding (GO: 0032555) were the first five pathways with the highest enrichment coefficient. In combination with the EvsF enrichment results, it was confirmed that the nucleotide-binding reaction affected plant disease resistance.

#### 2.4.2. KEGG Enrichment Analysis

DEGs from different plant combinations were analyzed for KEGG enrichment and distributed in five major categories. The EvsF Group had the largest number of differential genes and the most enriched pathways. EvsF, EvsG and FvsG are three common enriched pathway categories, including cellular processes, environmental information processing, genetic information processing and organismal systems. EvsF and EvsG are also enriched in metabolism pathways. The three pathways that are commonly enriched in comparison combinations include environment adaptation, membrane transport, translation, biosynthesis of other secondary metabolites, lipid metabolism, and amino acid metabolism, respectively, carbohydrate metabolism, metabolism of other amino acids, and metabolism of terpenes and polyketides. In addition, EvsF and EvsG are also differentially expressed in terms of energy metabolism, nucleotide metabolism, genetic information processing including folding, sorting and degradation, and transport and catabolism pathways.

KEGG functional enrichment analysis was performed for the three comparison groups, FDR values were calculated by hypothesis tests, and pathway enrichment significance was analyzed for the top 20 FDR values (Figure 4). In the EvsF combination (Figure 4a), the pathway is mainly concentrated in phenylpropanoid biosynthesis, starch and sucrose metabolism, MAPK signaling pathway-plant, carotenoid biosynthesis and isoquinoline alkaloid biosynthesis. Both EvsG and FvsG combinations (Figure 4b,c) were differentially gene-functionally enriched mainly in the plant–pathogen interaction pathway. The number of genes that differed among the comparison combinations was counted, and a Venn plot was drawn (Figure 4d). The results showed that there were 85 overlapping genes in the EvsF combination with the EvsG combination and 36 common genes in the EvsG combination with the FvsG combination; these shared genes were identified as key candidate genes regulating plant disease resistance.

There were 58 annotated genes in the combination of EvsF and EvsG; it encoded chitinase-like enzyme, beta-carotene 3-hydroxylase (CRTZ), acetyl-CoA carboxylase (ACC), LRR receptor-like serine/threonine protein kinase (FLS2, ERECTA and AT1G56130 and GSO1), wall-associated receptor kinase, antirust kinase (LR10), senescence-associated carboxylesterase, g-lectin s-receptor-like serine/threonine protein kinase (RKS1, LECRK4 and LECRK1), glutathione s-transferase (GST), and Mannan-1, 4-β-mannosidase (MAN), aspartic protease (AT5G10770), E3 ubiquitin protein ligase (RGLG2) and other functional enzymes. The results of KEGG enrichment analysis for common differential genes showed (Table 2) that were enriched on 12 pathways, affecting carbohydrate metabolism (fructose and mannose metabolism, amino sugars and nucleotide sugars metabolism), amino acid metabolism (tryptophan and glutathione metabolism), lipid metabolism, other secondary metabolites metabolism, plant signaling pathway and plant–pathogen interaction process.

A total of 27 genes were annotated in the combination of EvsG and FvsG; it encoded cellulose synthetase, ubiquitin-like specific protease, ω-hydroxypalmitate o-ferulyl transferase (HHT1), jasmonate o-methyltransferase, receptor-like protein kinase, E3 ubiquitin ligase (PQT3) and anthocyanin 3-O-glycosyltransferase. KEGG enrichment analysis of common differential genes (Table 2), mainly on six pathways, affected carbohydrate metabolism, lipid metabolism and secondary metabolite biosynthesis.

### 2.5. Expression Analysis of Disease Resistance Genes in Hybrid Populations

There are many resistance mechanisms in plants, such as tissue structure resistance and physiological and biochemical resistance. Plant cell surface or internal structure mainly through the formation of physical barriers to prevent the infection and further spread of pathogens, and produce some chemicals to inhibit the colonization of pathogens. In the above three groups (EvsF, EvsG and FvsG), GO enrichment and KEGG enrichment analysis were carried out for each comparison group, and the intersection of EvsF and EvsG, EvsG and FvsG were taken, respectively, to screen a number of differential genes. Based on this difference, this paper will analyze the expression of disease-resistance genes in the intraspecific crossing population of *V. vinifera* from the aspects of secondary metabolites, signal transduction and environmental adaptation, carbohydrates and transcription factors.

#### 2.5.1. Synthesis of Secondary Metabolites

Among the secondary metabolite synthesis-related genes, CRTZ of the CRT gene family showed differential expression in both EvsF and FvsG combinations (Figure 5). CRTZ expression was significantly higher in group E than in the other two groups. The expression level of the CRTZ gene in S1022 was about 15 times that in S1023. The expression level of CRTZ gene in the F and G groups was about the same. The POD gene was expressed in E, F and G, and the expression level of the POD gene in group E was significantly higher than that in the other two groups, and the expression level of S1022 in group E was about twice as high as that in group S1023. The POD expression levels of S1024, S1035, S1031 and S1037 were low. UDP-glycosyltransferase was the common differential gene between EvsG and FvsG. According to the difference in UGT gene expression between the two groups, there was no UGT gene expression between E group and F group—this gene was only identified in S1037 in group G.

#### 2.5.2. Signal Transduction and Environmental Adaptation

Among the genes associated with signal transduction pathways and environmental adaptation, there were four genes encoding FLS2 (VIT_00021648001, VIT_00002253001, VIT_00021648001 and VIT_00002253001 were named, respectively, FLS2-1, FLS2-2, FLS2-3 and FLS2-4) that were differentially expressed in different combinations (Figure 6a). The expression level of the four FLS2 genes was the highest in the S1022 and S1023 samples and lower in S1024, S1035 and S1037, but only slightly less in S1031. The expression of the CHI gene in the immune group (S1022, S1023) was the lowest compared with the high-resistant group (S1024, S1035) and the resistant group (S1031, S1037) (Figure 6a). There are seven genes encoding RPS2 (novel.14505, VIT_00023182001, novel.14509, VIT_00023173001, novel.14523, VIT_00023210001 and VIT_00023163001 were named respectively RPS2-1, RPS2-2, RPS2-3, RPS2-4, RPS2-5, RPS2-6 and RPS2-7), the expression levels of RPS2-3 and RPS2-5 were higher in the RPS2 family than in the rest of the RPS2 genes (Figure 6b), were not expressed in group G (S1031 and S1037), and were slightly higher overall in the high-antibody group than in the immune group.

#### 2.5.3. Carbohydrates

Four differentially expressed genes (VIT_000189230011, novel.73766, VIT_00007190001 and VIT_00004774001) related to carbohydrate anabolism were identified in this study. The expression levels of genes in groups F and G were higher than those in group E (Figure 7). Only novel.7376 was relatively high in group E, and VIT_00004774001 was only expressed in S1037 in the resistant group.

#### 2.5.4. Transcription Factors

The transcription factors identified in this study include AP2/EREBP, MYB, WRKY and so on. The transcription factors of the major families were analyzed. Seven WRKY transcription factors (VIT_00028244001, VIT_00035426001, VIT_00027069001, VIT_00030174001, VIT_00032661001, VIT_00035885001 and VIT_00022245001, were named, respectively, WRKY-1, WRKY-2, WRKY-3, WRKY-4, WRKY-5, WRKY-6 and WRKY-7) were detected, WRKY transcription factor gene expression in two samples from group E was higher overall than that in the other two groups (Figure 8a). VIT_00017315001, VIT_00002262001, VIT_00002262001 and novel.13847 (were named respectively AP2-EREBP-1, AP2-EREBP-2, AP2-EREBP-3 and AP2-EREBP-4) were identified in the EvsF comparison group, both samples in combination E exhibited high expression of these four transcription factors compared to group F (Figure 8b). Transcription factors with four MYB families (were named, respectively, MYB-1, MYB-2, MYB-3 and MYB-4) were also detected (Figure 8c), as was the case with the AP2-EREBP family. The transcription factor genes of this family were not detected in group G but expressed only in group E and group F, and the expression level of group F was significantly higher than that of group E.

## 3. Discussion

### 3.1. Evaluation of Disease Resistance in V. vinifera Hybrid Population

According to the analysis of the downy mildew disease index identification results, the disease resistance grades of the hybrid parents *Ecolly* and *Dunkelfelder* are, respectively, disease resistance and high resistance, disease resistance and disease resistance for two consecutive years, which is not completely consistent with the previous research results. The identification results of Wang et al. on six *V. Vinifera* downy mildew show that *Ecolly* and *Dunkelfelder* are, respectively, disease resistance and sensitive [22]. The reason for this difference may be related to the rainfall in the peak period of downy mildew in the experimental site. The average temperature in July, August and September in 2021 and 2022 in the Yangling area is concentrated at 20–30 °C, but the rainfall in two years is quite different, and sporangia needs to germinate in water. The relative humidity of leaves and the duration of wetting are crucial to the infection of pathogens [23]; therefore, there is a significant difference in the disease index during the peak period of onset between different years. In this study, through the identification of the disease resistance of the *V. vinifera* hybrid population, it was found that there were a certain number of hybrid individual plants among the populations with disease resistance grades superior to the hybrid parents. The overall performance was normal distribution, indicating that there was a super parental resistance inheritance in the *V. vinifera* hybridization. The resistance to downy mildew was continuously distributed in the *V. vinifera* hybrid population, belonging to the quantitative trait inheritance controlled by the European gene. This is consistent with previous research results and once again confirms that *V. vinifera* hybridization can breed new types that surpass disease-resistant parents. Choosing varieties with higher disease resistance as intermediate materials for breeding can accelerate the process of disease-resistance breeding [9,24]. The existence of superparent inheritance in the offspring population of *V. vinifera* hybridization exists in the grape varieties of *V. vinifera*. Micro-effective disease resistance multi-genes are exchanged and replaced among genes through intraspecific hybridization, and recurrent hybridization continuously introduces breeding materials with superparent traits, becoming an effective method for high-quality disease-resistance breeding of grapes [25,26]. However, the traditional *V. vinifera* intraspecific recurrent selection method is a relatively long process, and the introduction of molecular breeding can accelerate the process of selecting high-quality, disease-resistant new varieties using the *V. vinifera* intraspecific recurrent selection method. This also provides a good entry point for this study and exploring the micro-effective disease-resistant multi-genes in *V. vinifera* grape can provide a research foundation for developing molecular markers of micro-effective disease-resistant genes.

### 3.2. Disease Resistance and Secondary Metabolites

Plants can resist pathogens through multi-level defense strategies [27], in which secondary metabolites have an important impact on plant disease resistance [28]. Plant secondary metabolites such as terpenes, phenolics and nitrogen-containing compounds have a wide range of functions in plant disease resistance responses [29]. Terpenes can hinder the invasion of bacterial, fungal and various viral pathogens [30]. Carotenoid is a kind of yellow, orange-red or red pigment in higher plants and animals. Boba based their analysis of fusarium susceptibility on transgenic flax that produces excessive carotenoid and found that transgenic flax showed greater resistance to pathogen infection [31]. In order to increase the protein content, carotenoid content and disease resistance of wheat, we overexpressed the y and GPCB1 genes to increase the carotenoid content and protein content of wheat overexpression of these two genes also increased wheat resistance to leaf rust linkage reaction [32]. Alleles of the DN1 and DN2 genes with resistance to wheat aphids by Heng-Moss were affected by carotenoid concentrations [33]. Yu et al., through transcriptomic and proteomic analyses, found that carotenoid is particularly active in maize varieties resistant to gray leaf spot (GLS) and that carotenoid may enhance the defense mechanism of resistant varieties [34]. Carotenoid and chlorophyll contents in the samples treated with the pathogen were significantly increased in the test of lettuce resistance to *Botrytis cinerea* and *Sclerotinia sclerotiorum*; these two substances have beneficial effects on plant resistance [35]. “Nagasaki Kogane” potatoes with higher carotenoid levels were resistant to wilt [36]. Fraser studied the stable inheritance of the CRTB gene and its effect on carotenoid content. Specific expression of plant line synthase (CRTB) can increase carotenoid content in mature tomato fruit [37]. Using three genes encoding phytoene synthase (CRTB), phytoene desaturase (CRTL) and Lycopene β-cyclase (CRTY), extensive regulation of endogenous gene expression in carotenoid biosynthesis was observed in transgenic potato lines [38]. Maize endosperm overexpressed CRTB and CRTL increased carotenoid content in maize by 34-fold compared with that in non-overexpressing treatments [39]. The expression of β-carotenoid response transcription factor genes in CRTB transgenic soybeans was significantly different [40]. Co-expression of deoxy-d-xylose-5-phosphate synthase (DXS) and crtB increased β-carotenoid concentrations in cassava storage roots [41]. In this study, CRTZ of the CRT gene family was differentially expressed in both EvsF and FvsG combinations (Figure 6), and CRTZ expression was significantly higher in group E than in the other two groups; the expression level of the CRTZ gene in S1022 was about 15 times that in S1023. The expression level of the CRTZ gene in Group F and Group G was about the same, which indicated that the CRTZ gene could be used as a key candidate gene to improve disease resistance of grapevine as a potential gene for β-carotenoid biosynthesis; CRTZ is significant for screening disease resistance genes in grape hybrid populations.

Plant defense mechanisms are also significantly related to the accumulation of secondary metabolites, such as phenylalanine biosynthesis, which is regulated by multiple gene families; they included phenylalanine ammonia-lyase (PAL), cinnamic acid 4-hydroxylase (C4H), cinnamyl alcohol dehydrogenase (CAD), 4-coumaryl-coa ligase (4CL), peroxidase (POD), polyphenol oxidase (PPO), O-methyltransferase, hydroxycinnamyl transferase and caffeic acid 3-O-methyltransferase (COMT), PAL was a key enzyme in the phenylpropane synthesis pathway that encoded the biosynthesis of related plant defense hormones [42]. Plant hormone abscisic acid (ABA) was an important regulator of plant growth and stress resistance. The activities of PAL, PPO and POD were increased in the leaves of tomatoes treated with exogenous ABA; this, in turn, enhanced tomato resistance to pathogens [43]. Chen showed that the change in citrus disease resistance could be induced by regulating the biosynthesis of phenylalanine; transcriptome analysis showed that PAL, 4CL, C4H and POD genes were up-regulated after treatment [44]. POD was involved in many plant disease defense responses [45].

In addition, the relationship between BTH-induced disease resistance and the expression of PPO and POD genes, as well as the content of total phenolic compounds (TPC) in mango fruit, was also studied; the enhanced expression of these two genes played an important role in the BTH-activated mango defense response [46], and this activating effect was also validated in the effect of fusarium wilt on banana [47]. Phenylalanine biosynthesis was also involved in the natural defense of apples against various diseases. By inducing phenylalanine biosynthesis, the fruit can form a defense mechanism before the attack of pathogens. Zhao used transcriptomic and metabolomic analyses to show that genes involved in phenylpropane and phenylalanine biosynthesis were up-regulated, and the corresponding metabolite contents were increased, thus improving the disease resistance of fruits [48]. Phenylalanine biosynthesis also plays an important role in the resistance of sesame plants to fusarium oxysporum (Fos) [49]. Studies on plant–pathogen interactions have shown that PAL affects the accumulation of plant salicylic acid (SA) and phenolics. PAL activity was closely related to chlorogenic acid, the main component of total classified compounds [50]. Tobacco plants overexpressing phenylalanine ammonia-lyase PAL produced high chlorogenic acid (CGA) levels and were significantly less susceptible to fungal pathogen infection [51]. SA was involved in the defense of *Arabidopsis thaliana* leaves against injury. The activity of PAL and the expression level of SA mRNA in *Arabidopsis thaliana* leaves were significantly increased after ozone treatment [52], exogenous SA treatment reduced the accumulation of sugarcane mosaic virus (SCMV) and enhanced disease resistance in maize [53]. 

The inhibition of *Pseudomonas syringae* on *Botrytis cinerea* during grape storage was studied. The results showed that the cell suspension of *Pseudomonas syringae* could significantly inhibit the formation of *Botrytis cinerea* spores and significantly reduce the occurrence of *Botrytis cinerea*. In addition, the activities of PPO, POD, CAT, PAL, CHI and GLU increased in the fruits inoculated with Pseudomonas fluorescens suspension, which indicated that the resistance of the host was related to the activities of the enzymes. According to Riseh, when identifying the resistance of several wheat genotypes to take-all disease, it was found that different genotypes of wheat had different responses to take-all disease; the levels of PPO, PAL and total protein in the resistant group were higher than those in the susceptible group [54]. Multi-factor analysis indicated that POD might indirectly affect resistance, so POD could be used as a potential gene for disease resistance in grapevine.

Glycosyltransferase (GT) is found in almost all organisms and can transfer glycogroups from the activated donor molecule to the receptor molecule. The most common glycosyl donor in plants is UDP-glucose; the required glycosyltransferase is UDP-glucose transferase (UGT). Glycosyltransferase plays an important role in the synthesis of secondary metabolites, detoxification of endogenous and exogenous toxins, and plant growth and development. Among them, SA glycosylation was catalyzed by UGT. Tezuka et al. identified that UGT74J1 was expressed during plant development. The results showed that UGT74J1 was the key enzyme for SA accumulation in rice, and the resistance of over-expressed gene mutants to rice blast was also increased [55]. Pepper showed a hypersensitive effect to tobacco mosaic virus (TMV) infection. During the reaction of resistance to TMV, the UGT gene was up-regulated, while SA concentration in pepper decreased when the UGT gene was down-regulated; it is suggested that the UGT gene is involved in the TMV resistance response through the accumulation of SA [56]. SA carboxyl glycosyltransferase also regulates SA homeostasis in *Arabidopsis thaliana* through different mechanisms, playing a beneficial role in plant disease resistance [57]. It is suggested that UGT is only an enzyme involved in the process of SA accumulation, but SA accumulation depends on other mechanisms. It is suggested that UGT in this study is a minor gene controlling disease resistance.

### 3.3. Disease Resistance, Signal Transduction, and Environmental Adaptation

Plants recognize microbial invasions by detecting pathogen-associated molecular patterns (PAMP) through a pattern recognition receptor (PRR) [58]. Cell-surface-localized receptor kinases such as FLS2, EFR and CERK1 play critical roles in the detection of invading pathogens [59,60]. Flagellin2 (FLS2) is a leucine-rich repeat/transmembrane domain/protein kinase in the model plant *Arabidopsis thaliana*; the structure of protein receptor kinase is divided into an extracellular recognition domain, a single transmembrane domain and an intracellular kinase domain [61]. The extracellular recognition domain is responsible for recognizing the foreign signal molecule, then activating the receptor, causing the phosphorylation of the intracellular kinase domain, thus transmitting the signal from the extracellular to the intracellular, causing a series of biochemical reactions within the cell. FLS2 protein is a major receptor for bacterial flagellin recognition in plants. The overexpression of the FLS2 protein initiates immunity from microbial-associated molecular patterns, thereby inhibiting pathogen infection [62] and improving plant tolerance to bacteria, obtaining disease-resistant new plants. Activation of FLS2 does not involve its constitutive or ligand-dependent homodimer, but rather its binding to other proteins [63], binding of the highly conserved N-terminal epitope of FLS2 (flg22) induces the heteropolymerization and mutual activation of FLS2 with receptor-like kinase (Bak1), followed by plant immunity [64]. Yang demonstrated that the Arabidopsis osteoporosis-like protein (ORM), a negative regulator of sphingolipid biosynthesis, acts as a selective autophagy receptor to mediate FLS degradation, that is, plants overexpressing ORM1 or Orm2 are undetectable or substantially reduce FLS2 accumulation [65]. Thus, if FLS is to be highly expressed, the intervention of Orm should be excluded from the plant genetic system. This study confirms the critical role of FLS proteins in plant defense mechanisms against pathogen infection. In general, flg22 of the intraspecific crossing population of *V. vinifera* could induce the heterogenization and interaction between FLS2 and BAKI, and the protein could recognize the pathogen and initiate plant immunity.

Chalcone isomerase/chitinase (CHI) is a key enzyme in flavonoid biosynthesis, which determines the production of flavonoid compounds in plants. In plants, one coumaryl-coa and three malonyl-coa are activated by chalcone synthase (CHS) to produce chalcone, which is then subjected to CHI to produce dihydroflavones. In turn, various types of flavonoid and flavonoid compounds are produced. The results showed that tomato plants treated with CHI had resistance to wilt caused by *Ralstonia solanacearum*. The defense genes of the tumor protein homologs involved in the translational control of plant stress were highly up-regulated in the treatment, and treatment reduced disease incidence by 56.6% [66]. In studying the properties of chitinase and its antifungal activity in Celestine grapes, it was found that the chitinases (CHI-LA, CHI-1B and CHI-2) purified from the active part of the chitin protein band were related to the inhibition of grapes on the growth of *Botrytis cinerea* mycelium [67]. However, in this study, the expression of the CHI gene in the immune group (S1022 and S1023) was lower than that in the resistant group (S1024 and S1035) and the Resistant Group (S1031 and S1037) (Figure 8a). In a systematic study by Chen to understand the biochemical mechanisms of *Fusarium graminearum* (FG) resistance, differences in plant metabolome associated with FG resistance between transgenic *Arabidopsis thaliana* expressing CHI, fusarium-specific recombinant antibody gene (CWP2) and transgenic *Arabidopsis thaliana* co-expressing CHI-CWP2. It was found that *Arabidopsis thaliana* expressing CHI-CWP2 showed higher FG resistance, lower disease index and lower levels of mycotoxins compared with wild type and plants expressing CHI and CWP2 alone [68]. It may be that CHI expression needs to interact with other proteins to activate in order to exert disease resistance.

The plant R protein, which recognizes nontoxic effectors directly or indirectly, only functions when activated by the corresponding effectors. Finding the key effectors can help identify the corresponding R genes and apply them to disease-resistant breeding. At present, the Rps gene is considered to be the most effective method to control *Phytophthora sojae*. More than 30 Rps genes have been identified on nine chromosomes. Among them, chromosome 3 has the most Rps genes, Rps1 (including five alleles: Rps1A, Rps1B, Rps1C, Rps1D and Rps1K) [69], Rps7 [70], RpsYD25 [71], Rps9 [72], RpsYU25 [73], RpsYD29 [74], RpsUN1 [75], RpsAH, RpsZheng, RpsQ [76], RpsWY [77], RpsHN [78], RpsHC18, RpsX [79], RpsGZ and Rps14 [80], and Rps1498. In addition, Rps3 (including the three alleles Rps3A, Rps3B and Rps3C) and Rpssn10 were mapped on chromosome 13 and linked to Rps8; Rps2 and RpsUN2 were located on chromosome 16; Rps4, Rps5, Rps6, Rps12, RpsJS and Rps13 are located on chromosome 18 [81] and linked to Rps12 and Rps13, respectively. Rps2 is a single-copy resistance gene of *Arabidopsis thaliana*, which mediates resistance to bacterial pathogens carrying the AVRRPT2 gene. The results showed that the amino acid sequence of Rps2 contained leucine repeats, transmembrane domain, leucine-linked domain and p-loop domain [82]. However, when the bacterial effector protein AvrRpt2 was delivered to *Pseudomonas syringae* to induce resistance in Arabidopsis plants expressing Rps2, the interaction between Rps2 and Arabidopsis RIN4 was found, avrrpt2 leads to inactivation of RIN4 during Rps2 activation. Avrrpt2-mediated inactivation of RIN4 occurs in the context of Rps2 mutations, suggesting that Rps2 initiates signaling based on the effect of inactivation of RIN4 rather than direct recognition of Avrrpt2 (2003). In this study, Rps2-3 and Rps2-5 were more highly expressed than the other Rps2 genes (Figure 8b). They were not expressed in group G (S1031, S1037) but were slightly higher in the middle resistant group as a whole. Rps2 needs the activation of regulatory factors and inactivation of other substances, so the Rps2 gene can be used as an auxiliary gene to explore disease resistance genes in grapes in combination with other relevant regulators to exert its function.

### 3.4. Disease Resistance and Carbohydrates

Plants usually respond to biotic or abiotic stresses by altering components, such as cell wall components and metabolites. The cell wall is one of the first lines of defense to protect cells from the invasion of fungal pathogens, and it is also the main basic resistance barrier of the host. Plant cells adjust their cell walls to withstand the physical and chemical forces exerted by the fungus in order to prevent penetration [83]. Daniele compared the different levels of horizontal resistance to late blight in two potato hybrids. The carbohydrate contents in stems and leaves of infected and uninfected plants were determined by high-performance liquid chromatography. Some carbohydrates accumulate in the stems of infected resistant hybrids but remain constant in susceptible hybrids. In addition, these carbohydrates accumulate in leaves only in susceptible hybrids that are infected. Szugyi established the relationship between disease resistance and carbohydrate content in cherry plant tissues. The results showed that the glucose content, the ratio of glucose and fructose to sucrose of sour cherry varieties and their hybrids were related to disease resistance. The susceptible genotype had higher glucose content, and the ratio of hexose to sucrose was significantly higher than that of the resistant genotype [16].

Polyol/monosaccharide transporters (pmts) are involved in phloem loading and unloading in sugar transport [84]. Mannitol is a six-carbon acyclic sugar solution. In plants, mannose-d-mannopyranose 6-phosphate is integrated into mannitol-1-phosphate and then dephosphorylated to mannitol. In plant species, mannitol is found in more than 70 families [85] and has functions as a carbon storage compound, a reducing force reservoir [86], a compatible osmotic agent, and osmoregulation. Mannitol can also abrogate reactive oxygen species (ROS) [87,88], presumably to assume a cell-reinforcing function in host-pathogen interactions, and plants use ROS as an antimicrobial signaling molecule to initiate different protective responses. Studies have also shown that mannitol and mannitol dehydrogenase (MTD) are involved in the protection of plant pathogens. Three non-mannitol plants (tobacco, tomato and Arabidopsis) have been found to contain pathogen-induced MTD; among them, mannitol may act as a signaling molecule [89,90]. Fungal infestation in plants reduces endogenous levels of free sugars, amino acids, nucleosides, and organic acids [91]. In this study, no significant resistance genes were found in the pathway related to carbohydrate synthesis, presumably related to the transcriptomic analysis of the samples under natural rather than stress conditions; signal molecules need to be triggered by pathogens.

### 3.5. Disease Resistance and Transcription Factor Regulation

Transcriptional regulators are a class of proteins that regulate gene expression in organisms and are critical steps in the response to stress [92]. It is reported that about 2296 transcription factors belonging to 58 transcription factor families have been identified in *Arabidopsis thaliana*; the AP2/EREBP family, NAC family, WRKY family, bHLH family, MYB family, bZIP family and homeodomain proteins are involved in plant disease resistance; among them, ERF, NAC, WRKY, MYB and bHLH are key transcription factors in signaling pathways [93].

Transcription factors can regulate the expression of several genes related to disease resistance traits and enhance the expression of disease resistance genes by enhancing the effects of several key regulators, thereby improving disease resistance in plants [94]. The transcription factors of the AP2/EREBP family have been reported to play a variety of roles throughout the life cycle of plants: from key regulators of several developmental processes to the generation of resistance responses to various stresses in plants [95]. Kumari identified 54 AP2/EREBP genes in maize and identified key regulators to improve stress adaptation and tolerance by physical mapping, gene structure, and conserved motifs. A large number of transcription factors have evolved to function in complex regulatory networks in plants, and Zeng et al. identified the interaction of 1970 AP2/EREBP proteins in *Arabidopsis thaliana* and revealed the evolutionary patterns of the family’s transcriptional genes in cruciferae [96]. One of these was identified to encode an AP2/Erebp-type transcription factor (designated the tobacco callus expression factor NTCEF1) that is highly expressed in various types of tobacco calli, including CHRK1 transgenic calli. NTCEF1-overexpressed arabidopsis plants showed enhanced resistance to the bacterial pathogen pseudomonas syringae. However, studies on the AP2/EREBP gene in grapes have been less reported. In this study, the four transcription factors were highly expressed in two samples of E combination compared with those of the F group, suggesting that the AP2/EREBP family is a key transcription factor regulating disease resistance-related genes in grapevine.

The first layer is PAPM-triggered immunity (PTI), which is activated by recognition between pathogen-associated molecular patterns and plant pattern recognition receptors; the second layer was triggered by the recognition of the pathogen effect by plant resistance protein (ETI). Bai identified 83 WRKY genes in tomatoes; these genes and their homologs have also been shown to respond to abiotic and biotic stresses in Arabidopsis and rice [97]. PTI and ETI accept regulation of WRKY at different regulatory levels, respectively, and WRKY transcription factor-complex family members are involved in the regulation of transcriptional reprogramming associated with plant immune responses. WRKY transcription factor induces flavonoid accumulation in cotton by up-regulating the expression of several genes involved in flavonoid biosynthesis. In this process, the WRKY-mitogen-activated protein kinase (MAPK) cascade promotes flavonoid biosynthesis to defend against pathogen infection [98]. Wei performed transcriptional analysis on rice in response to the early infection of Magnaporthe grisea and verified that overexpression of WRKY47 conferred resistance to the fungus through complex coding involving signaling and metabolic pathways [99]. The expression of the WRKY1 gene is regulated by developmental mode, salicylic acid, ethylene and hydrogen peroxide in berries and leaves; the overexpression of grape WRKY1 in tobacco reduced susceptibility to various fungi but not to viruses [100]. Heterologous overexpression of WRKY53 isolated from Chinese wild grapes in *Arabidopsis thaliana* can accelerate leaf senescence and improve disease resistance [22]. In this study, the WRKY transcription factor gene expression level of two samples in group E was higher than that of the other two groups in EvsF, EvsG and FvsG, but VIT-00030174001 showed the highest expression level in group F.

Transcriptional regulation of host cells plays an important role in establishing plant defense and associated cell death in response to pathogen attack, and current research on transcriptional regulation of plant defense focuses on the MYB family of transcription factors [101]. The expression of the MYB1 gene was induced during the defense response of tobacco to TMV and P. syringae, and the MYB1 protein bound to the promoter of defense-related genes, suggesting that it plays a role in the regulation of the defense response [102]. Liu found that cotton with the MYB36 gene silencing was more sensitive to drought stress and verticillium wilt than cotton without MYB36 gene silencing, transient expression of fused MYB36-GFP in tobacco cells was able to localize MYB36 to the nucleus, and MYB36 functions as a transcription factor by enhancing the expression of related genes, involved in drought tolerance and verticillium wilt in Arabidopsis and cotton [94]. 

*Botrytis cinerea* is a fungus that infects grapes. The interaction between Erf16 and MYB306, a member of the ethylene response factor (ERF) family, has been reported. The overexpression of ERF16 and MYB306, as well as overexpression of the ERF16-MYB306 transcription complex, increases resistance to *S. cinerea* in grape leaves, whereas the silencing of either gene leads to resistance to S. cinerea [103]. Li screened transcription factors related to grapevine downy mildew, including ERF, MYB, WRKY and bHLH. MYB44, isolated from grapes, was significantly induced in response to the defense expressed in grapes after *S. cinerea* infection, interacting with the salicylic acid-responsive transcriptional coagulant NPR1 for defense expression [104]. In this study, the expression levels of transcription factor genes of the MYB family were also different among different samples, but the expression levels of the high-resistance group were significantly higher than those of the immune group. Although the transcription factors of the MYB family were beneficial to the regulation of plant disease resistance, they need to combine with other key proteins or genes.

## 4. Methods and Materials

### 4.1. Experimental Materials

The high-quality disease-resistant variety *Ecolly* [*Chenin Blanc* × (*Chardonnay* × *Riesling*)] × (*Baishinan*, *Chardonnay*, *Riesling*) through recurrent hybridization within *V. vinifera* and high-quality variety *Dunkelfelder* (Anjiwen × Ziye) and 37 hybrid seedings (*Ecolly* × *Dunkelfelder*) were planted in the vineyard of Shengtang Winery in Yangling, Shaanxi province (lat. 34° N, long. 108° E) in 2020. The plants were controlled under similar management conditions such as soil, irrigation, pruning and disease control.

### 4.2. Identification of Resistance to Downy Mildew

During the peak period of grape downy mildew, fresh leaves were collected, and new sporangia were grown on the diseased leaves. The sporangia were gently brushed into sterile distilled water with a sterile soft brush and shaken well. The number of sporangium was calculated by the blood cell counting board, and the concentration of sporangium suspension was adjusted to 1 × 10^5^ cells/mL.

This study selected six intact leaves from each test plant, washed and dried them, and cut 30 circular plates using a 15 mm inner diameter punch. The researcher placed the leaf circular plates with the back facing upwards into a 4 mL sterile distilled water culture dish and randomly placed 10 leaf circular plates into each culture dish. Three technical replicates were set for each seed quality, and three biological replicates were set for hybrid parents. An inoculated 30 per disc μL (300 sporangia) sporangia suspension is placed in the center of a round leaf and placed in a culture chamber, with a relative humidity of 90%, 20 ± 2 °C, and alternating cultivation under light and darkness for 12 h. After 24 h of inoculation with the downy mildew pathogen, use sterile filter paper to remove the inoculation suspension from the leaf disc and continue cultivation under the same conditions. After 7 days (168 h) of inoculation, the leaf discs were photographed with digital camera, and the percentage of diseased area was calculated with Photoshop CS5 (Adobe, San Jose, CA, USA), according to the Desaymard “0.10” classification method, the percentage of the infected area on the leaf disc to the whole leaf disc was classified: Grade 0: no lesion; Grade 1: the infected area accounted for 0.1~5% of the whole leaf disc; Grade 3: the infected area accounted for 6~25% of the whole round leaf; Grade 5: the infected area accounted for 26~50% of the whole round leaf; Grade 7: the infected area accounted for 51~75% of the whole round leaf; and Grade 9: the infected area accounted for 76~100% of the whole round leaf. The disease index was calculated according to the following formula:$$\mathrm{DI}=\frac{\sum \left(\mathrm{Number}\;\mathrm{of}\;\mathrm{diseased}\;\mathrm{leaves}\;\mathrm{at}\;\mathrm{all}\;\mathrm{levels}\;\times \;\mathrm{Corresponding}\;\mathrm{disease}\;\mathrm{level}\right)}{\mathrm{Survey}\;\mathrm{total}\;\mathrm{number}\;\mathrm{of}\;\mathrm{leaves}\;\times \;\mathrm{Highest}\;\mathrm{disease}\;\mathrm{level}}\times 100$$

According to the standard of the international plant germplasm committee (IBPGR), the infection degree of grape leaf discs was classified into five grades. Grade 1: Disease Index 0, disease resistance is immune; Grade 2: Disease Index 0.1–5.0, disease resistance is high; Grade 3: Disease Index 5.1–25.0, disease resistance is disease resistance; Grade 4: Epidemic Index 25.1–50.0, the resistant degree is susceptible; and Grade 5: Epidemic Index 50.1–100, the resistant degree is highly susceptible.

### 4.3. Transcriptome Material

Based on the results of the disease resistance identification in vitro, two plants of different disease resistance types were selected as biological replicates, including immune, high resistance and disease resistance, under natural conditions (non-pathogen stress). Young tissues were collected for transcriptome sequencing to screen disease resistance-related genes.

### 4.4. Transcriptome Sequencing

#### 4.4.1. RNA Extraction, Quality Control, Transcriptome Library Construction, and On-Line Sequencing

RNA was extracted from tissue samples, and the concentration, purity, and integrity of the RNA were detected.

The construction of the cDNA library would start after the quality of the sample was qualified. The main process is as follows:(1)Enrichment of total RNA using mRNA enrichment method: Enrichment of mRNA with polyA tail using magnetic beads with Oligo dT.(2)Fragmentation of the obtained RNA using interrupted buffers, reverse transcription with random N6 primers, and synthesis of cDNA double-stranded to form double-stranded DNA.(3)Flatten the ends of the synthesized double-stranded DNA and phosphorylate the 5′ end, forming a protruding “A” sticky end at the 3′ end, and then connect a bubbly junction with a protruding “T” at the 3′ end.(4)The connecting products were amplified by PCR using specific primers.(5)The PCR product is thermally denatured into a single strand and then cyclized with a bridge primer to obtain a single-stranded circular DNA library. The cDNA library was sequenced based on the technology of Sequencing while Synthesis (SBS).

##### Data Quality Control and Analysis

The original sequencing data will contain linker information, a low-quality base, and an undetected base (in the form of N), which will cause great interference with the subsequent information analysis. In order to eliminate the interference information, a fine method is needed, and the data obtained are effective data, namely clean reads.

The raw reads were filtered by Soapnuke [105] software (v2.1.0). The filtering steps were as follows:(1)Paired reads containing linker sequences were filtered.(2)The proportion of paired reads that remove n (N for undetermined base information) is greater than 0.5%.(3)Removal of low-quality paired reads.

Sequencing data output statistics: The results of sequencing image recognition, decontamination, to the joint. The statistical results included the number of sequenced reads, the data yield, the Q20 content, the Q30 content, the GC content and so on.

Using RSEM [106], the alignment results of Bowtie2 were analyzed, and the number of Reads Per sample aligned to each transcript was calculated and converted to FPKM (Fragments Per Kilobase per Million bases), paired-end reads from the same fragment were counted as a fragment to obtain the expression level of the gene and transcript.

#### 4.4.2. Analysis of Differentially Expressed Genes (DEGs)

(1)Normalization of the original read count, mainly to correct for sequencing depth.(2)Statistical models were calculated for hypothesis-testing probability (*p*-value).(3)Multiple hypothesis testing was performed to obtain the FDR (discovery error rate).

Based on the results of the difference analysis, we screened for genes that met the conditions of FDR < 0.05 and |log_2_FC| > 1 (FC, Fold change, Fold of difference) as significantly different genes.

The analysis of differential genes was carried out in three ways: (1) comparison between immune and highly resistant materials, (2) comparison between immune and resistant materials, and (3) comparison between highly resistant and resistant materials. In (1)(2), the difference genes of the two comparison groups were crossed, and in (2)(3), the same difference genes were crossed.

#### 4.4.3. Functional Annotation and Enrichment Analysis of Differential Genes

Gene function annotation and significant enrichment analysis were performed in the GO and KEGG databases [107]. Trend analysis was performed with ShortTime-series Expression Miner [108] software, with a file inputting the expression amounts of all differential genes of the grouped samples in which the samples were grouped (arranged in the order of the samples according to biological logic). Then, the KEGG/GO functional enrichment analysis was performed for the genes in each trend, and the *p*-value was calculated by a hypothesis test. After FDR correction, the threshold of the Q-value ≤ 0.05 is defined as KEGG/GO, which is significantly enriched in this trend.

### 4.5. Data Analysis

The original data were recorded, sorted and analyzed by Microsoft Excel 2021, the significance analysis was carried out by SPSS, the Logistic equation was fitted by Origin Pro 2021, and the graph was drawn.

## 5. Conclusions

There are differences in disease resistance among *V. vinifera* cultivars, which are controlled by minor effect polygenes. The results of transcriptome sequencing of materials with different disease resistance types in natural conditions showed that there were many different genes among different disease resistance samples; these differentially expressed genes were mainly enriched in the following GO items: metabolic process, cellular process, catalytic activity, binding reaction, cell anatomical entity, etc. Environmental adaptation, membrane transport, translation, biosynthesis of other secondary metabolites, lipid metabolism, amino acid metabolism, carbohydrate metabolism, metabolism of other amino acids and metabolism of terpenes and polyketones are common differential pathways of KEGG enrichment. CRTZ, COMT, POD and UGT genes in the secondary metabolite pathway, FLS, CHI, Rps2 family genes in signal transduction and environmental adaptation, WRKY, MYB and AP2-EREBP are the key candidate genes for exploring disease resistance in *V. vinifera*.

This study provides theoretical support for the cross-breeding of new *V. vinifera* cultivars with good quality and disease resistance at the molecular level; the common differential genes can be used to develop molecular markers and further validate the accumulation of substitutions of minor genes. Due to the influence of the number of hybrid populations, this study did not directly construct a genetic map but combined with the analysis of disease resistance phenotype and transcriptome to mine multi-genes of disease resistance. In the future, we will expand the number of hybrid populations, construct the high-density genetic map, and combine the disease-resistant phenotype and transcriptome analysis to continue mining disease-resistant multi-genes and systematically analyze the accumulation of substitution genes in *V. vinifera* intraspecific hybridization.

## Figures and Tables

**Figure 1 ijms-24-15311-f001:**
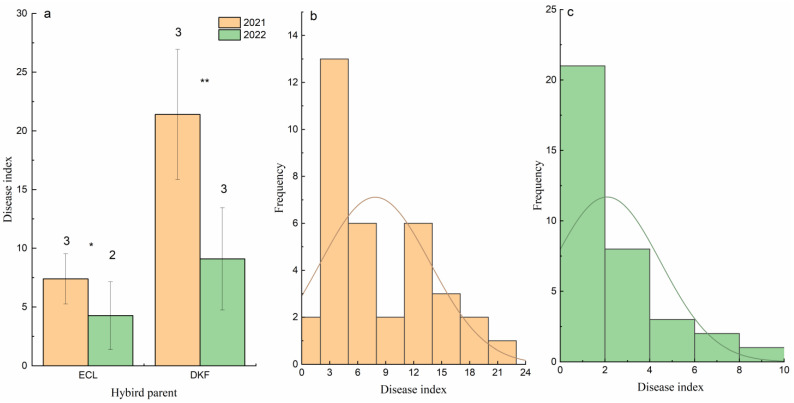
Disease index and genetic distribution of resistance to downy mildew in hybrid population. The disease index of (**a**) hybrid parent in 2021 and 2022, the genetic distribution of resistance to downy mildew in (**b**) hybrid population in 2021, and the genetic distribution of resistance to downy mildew in (**c**) hybrid population in 2022. Among them, ECL stands for *Ecolly,* and DKF stands for *Dunkelfelder*. * indicates significant difference (*p* < 0.05); ** indicates very significant difference (*p* < 0.01).

**Figure 2 ijms-24-15311-f002:**
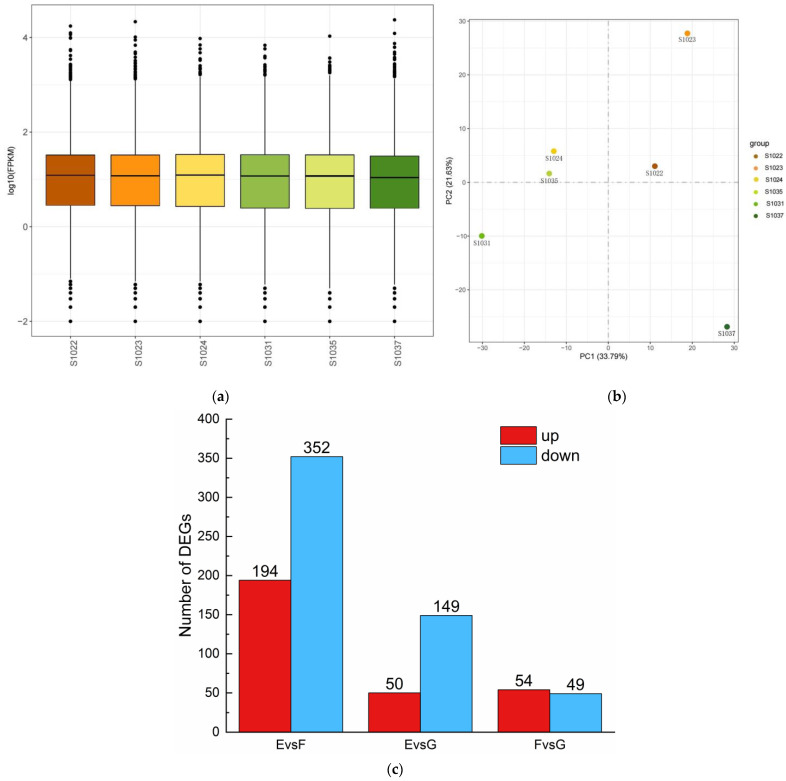
Distribution, characteristics, and up-and down-regulated genes of FPKM in all samples. (**a**) FPKM box plot of transcripts from different samples; (**b**) PCA plot of all samples; and (**c**) up-and down-regulated genes for each combination.

**Figure 3 ijms-24-15311-f003:**
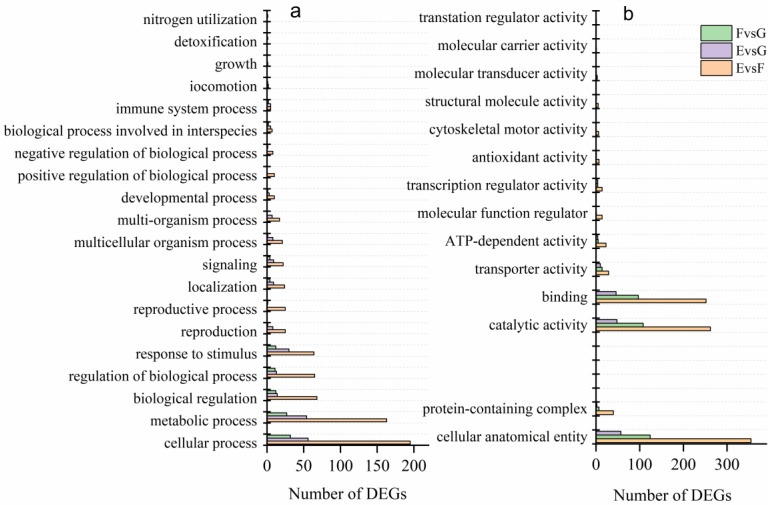
GO enrichment among different resistance combinations. (**a**) Biological process; (**b**) cell process and metabolic process.

**Figure 4 ijms-24-15311-f004:**
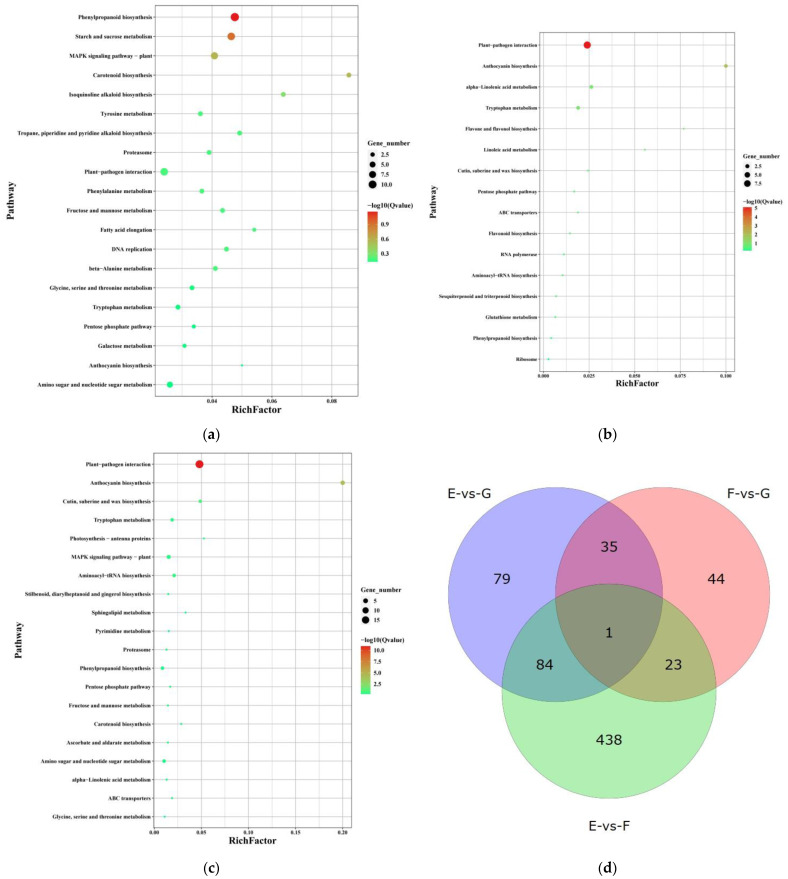
KEGG enrichment among different resistance combinations, (**a**) EvsF group, (**b**) FvsG group, (**c**) EvsG group, and (**d**) Venn diagram of different genes among different resistance combinations.

**Figure 5 ijms-24-15311-f005:**
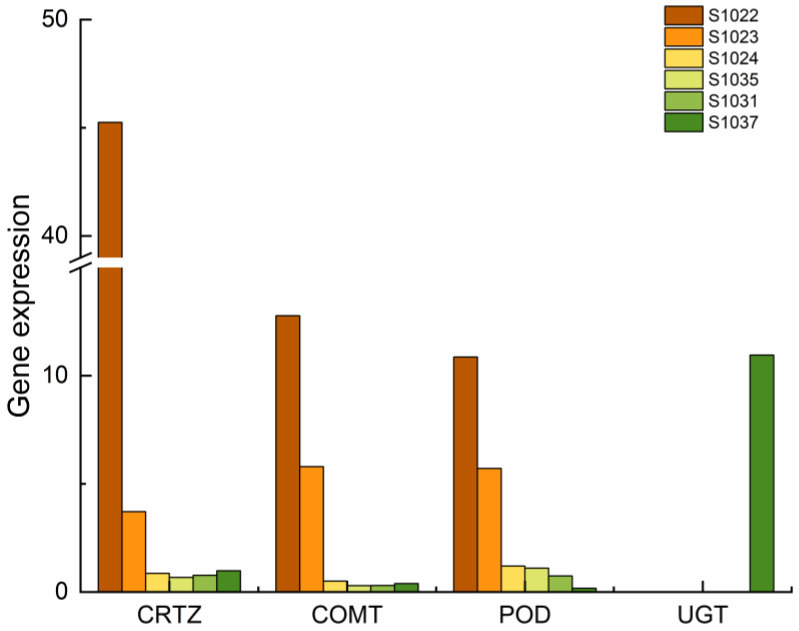
Expression levels of CRTZ, COMT, POD and UGT genes in plants.

**Figure 6 ijms-24-15311-f006:**
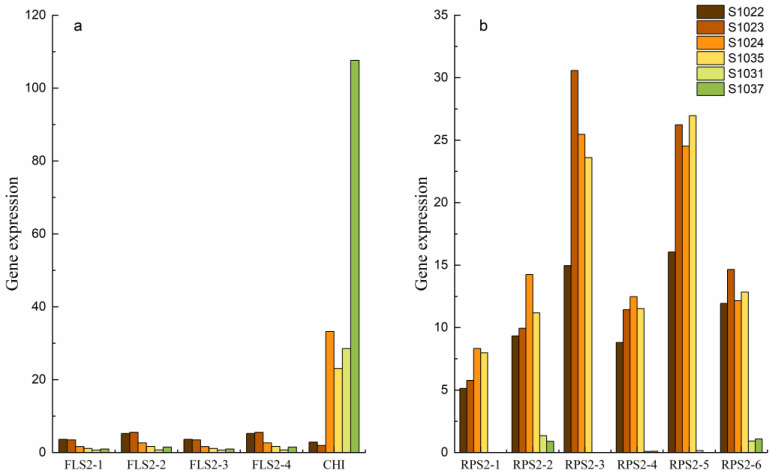
Expression levels of disease resistance-related genes, (**a**) FLS and Chi genes, and (**b**) BRPS2 genes.

**Figure 7 ijms-24-15311-f007:**
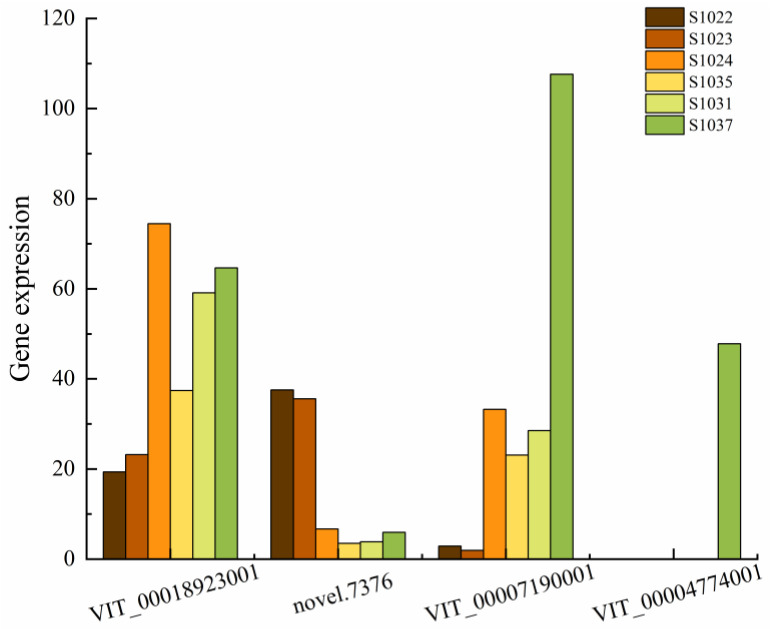
Carbohydrate-related gene expression amount.

**Figure 8 ijms-24-15311-f008:**
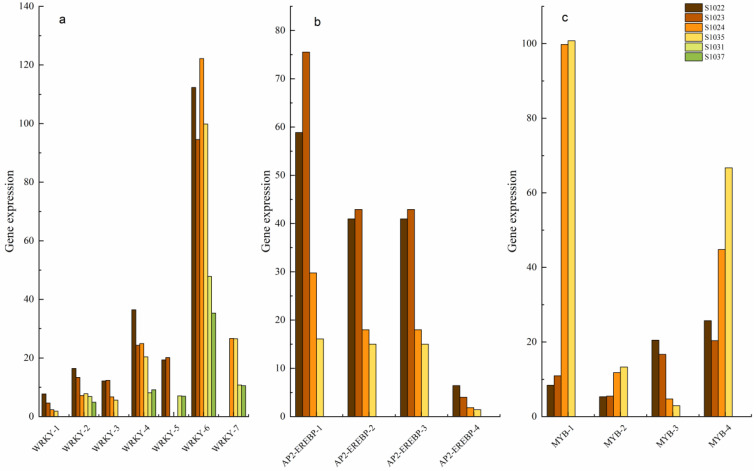
Expression level of transcription factor-related genes. (**a**) WRKY family gene expression; (**b**) AP2-EREBP family gene expression; and (**c**) MYB family gene expression.

**Table 1 ijms-24-15311-t001:** Statistics related to RNA detection.

Sample Name	Clean Reads Pairs	Clean Base (GP)	Length	Q20 (%)	Q30 (%)	GC (%)	Total Mapped Ratio%
S1022	26.14	7.84	150; 150	97.0; 97.3	91.1; 91.7	46.0;46.0	87.38
S1023	29.74	8.92	150; 150	96.9; 97.0	90.9; 90.7	46.0; 45.9	89.44
S1024	35.59	10.68	150; 150	96.8; 97.1	90.7; 91.1	45.8; 45.7	90.09
S1031	41.72	12.52	150; 150	97.0; 97.6	91.0; 92.3	45.7; 45.7	91.53
S1035	39.71	11.91	150; 150	97.2; 97.4	91.5; 91.9	46.0; 45.9	92.20
S1037	35.66	10.70	150; 150	97.1; 97.6	91.3; 92.3	46.2; 46.2	90.91

**Table 2 ijms-24-15311-t002:** KEGG enrichment analysis of common differential genes in all combinations.

	Classification	Access ID	Access	Genes	Functional Annotations
EvsF and EvsG	Signal transduction	ko04016	MAPK signaling pathway-plant	VIT_00021648001	FLS2; receptor protein kinase
				VIT_00007190001	CHI chalcone isomerase B; chitinase
				VIT_00002253001	FLS2; receptor protein kinase
	Environmental adaptation	ko04626	Plant–pathogen interactions	VIT_00021648001	FLS2; receptor protein kinase
				VIT_00002253001	FLS2; receptor protein kinase
	Amino acid metabolism	ko00380	Tryptophan metabolism	VIT_00034498001	COMT; catechol-o-methyltransferase; caffeic acid 3-O-methyltransferase
				VIT_00011005001	YUCCA; auxin synthesis-related enzyme
	Biosynthesis of other secondary metabolite	ko00940	Phenylalanine biosynthesis	VIT_00034498001	COMT; catechol-o-methyltransferase; caffeic acid 3-O-methyltransferase
				VIT_00015533001	Peroxidase (E1.11.1.7)
	Lipid metabolism	ko00600	Sphingomyelin metabolism	VIT_00033008001	SPT; long-chain base biosynthetic protein 2a subtype X2
	Carbohydrate metabolism	ko00051	Fructose and mannose metabolism	VIT_00018923001	MAN; mannan-end-1,4-β-mannosidase 7
		ko00520	Amino and nucleotide sugars metabolism	novel.7376	Class IV chitinase precursor; E3.2.1.14
				VIT_00007190001	CHIB
	Metabolism of terpenes and polyketones	ko00906	Carotenoid biosynthesis	VIT_00019403001	crtZ; Beta-carotene 3-hydroxylase 1, chloroplast
	Translation	ko00970	Aminoacyl tRNA biosynthesis	VIT_00020308001	FARSA, pheS
		ko03010	Ribosome	VIT_00030972001	RP-L26e, RPL26
	Metabolism of other amino acids	ko00480	Glutathione metabolism	novel.18953	GST, gst; Glutathione S-transferase
	Folding, classification, and degradation	ko03050	Proteasome	novel.21935	PSMD13, RPN9; putative protein CK203_114353
EvsG and FvsG	Biosynthesis of other secondary metabolite	ko00942	Anthocyanin biosynthesis	VIT_00037411001	UDP glycosyltransferase
	Environmental adaptation	ko04626	Plant–pathogen interactions	novel.14505	Disease-resistant protein RPS2
				VIT_00023182001	Resistance protein (RPS2) At4g27190
				novel.14509	Disease resistance protein (RPS2)
				VIT_00023173001	ADP binding RPS2
				novel.14523	Disease resistance protein (RPS2)
				VIT_00023210001	Putative protein RPS2
				VIT_00023163001	ADP in combination with RPS2
	Translation	ko00970	Aminoacyl tRNA biosynthesis	VIT_00030775001	Valine-tRNA ligase; mitochondria/chloroplasts (VARS, valS)
	Lipid metabolism	ko00073	Keratin, lutein and wax biosynthesis	VIT_00030078001	ω-hydroxypalmitate o-ferulate transferase (HHT1)
		ko00592	α-linolenic acid metabolism	VIT_00009751001	Jasmonate o-methyltransferase
	Carbohydrate metabolism	ko00030	Pentose phosphate pathway	VIT_00004774001	

## Data Availability

The datasets generated during and/or analyzed during the current study are available from the corresponding author upon reasonable request.

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
