# Peer review of "Mining of Minor Disease Resistance Genes in V. vinifera Grapes Based on Transcriptome"

_ijms, 2023, doi:10.3390/ijms242015311_

Round 1

Reviewer 1 Report

My remarks might fall short, as I am far from a transcriptomic expert. I am impressed by the extent of discussion based on 6 samples from two contrasted years for field trials and downy mildew resistance behavior.

My minor remarks are more about to facilitate the comprehension and the readiness of the figures. I leave to a transcriptomic expert the comments after §2.4.

Author Response

Dear reviewer, thank you very much for your suggestion. In response to your question, I will provide the following response:

(1)There is a significant difference in rainfall between 2021 and 2022 in the Yangling area, and there is a significant difference in the disease index of downy mildew during the peak period between years. However, the disease index ranking of hybrid parents and offspring is the same between the two years. The selected immune, high resistance, and medium resistance groups have the same resistance within two years.

(2)Single plants with stable resistance were selected from all samples for two consecutive years, and two single plants were selected from the immune group, high resistance group, and medium resistance group to screen key genes more accurately.

(3)The resolution and color of all images have been updated.

Reviewer 2 Report

This paper is an interesting findings by Liu et al., as it helps to looks for disease resistance  genes that may be important for Vitis vinifera breeding. How before the paper can be published, I have few minor comments: 

Line 20-21: It is not clear as to what comparisons were done and the results of the respective comparison. Can the authors please explain.

Line 131: It is not clear what authors mean by R2 in this line

Line 145-146: Can the authors explain what were the pairwise comparisons. It will not be clear to the readers from reading these line. Also what does 3 sets of transcripts mean here?

Figure 4: Figure panel number a, b,... should be listed on the top

Line 226-227: THe authors have discussed about the 58 annotated genes but its not clear if these 58 genes are out of 85 or 36 overlapped genes which they mentioned before.

Line 260-269: The results in these lines need to be explained better. It is not very clear what the authors want to state in these lines. Consistency between E, F, G and labelling of S1022-S1037 needs to clarified in the figure legends. 

Line 286: what is high-antibody group?

Methods

Line 650-654: The method section in these lines is very confusing in this section. Authors should clearly states how this part was done. As mentioned above, some more details about naming of S1002-S1037 should be described in the method section

Line 673: Authors should explain at least in brief the methods for RNA-seq library preparation. It would be useful for readers who would like to reproduce the study.

Some Additional Comments:

  1. The figures are poorly done and are of very poor quality. Authors need to improve the resolution. Also for figure 2, all the panels should come under one figure and not in two.
  2. Figured with Y-axis as gene expression should be clearly labelled what is the metrics for gene expression like FPKM or TPM
  3. Also, it would be important to confirm some of these genes by qPCR.
  4. Discussion is too long and needs to be shortened and made relevant for the results shown in the paper.

Text needs to be proof read as there are some typological errors. 

Author Response

Dear reviewer, thank you very much for your suggestion. In response to your question, I will provide the following response:

(1)Lines 20-21 were screened based on the resistance characteristics of hybrid populations to downy mildew. Two immune, two high resistance, and two medium resistance single plants were selected as combinations, and the differential genes between different resistance combinations were analyzed to obtain genes related to downy mildew resistance in vitis vinifera.

(2)R2 is an error identifier, correctly represented as R ²≥ 0.98.

(3)The three transcripts are the immune group, high antibody group, and medium antibody group. Pairwise comparison refers to the comparison between the immune group and the high antibody group, the immune group and the medium antibody group, and the high antibody group and the medium antibody group.

(4)The 58 genes mentioned in lines 226-227 are part of the 85 overlapping genes mentioned earlier, and only 58 genes were annotated during the analysis process.

(5)Lines 260 to 269 mainly describe the differential expression and expression levels of genes related to secondary metabolic pathways among different resistance combinations.

(6)High resistance refers to the high level of resistance to downy mildew.

(7)The resolution and color of all images have been updated.